# Nematicidal Activity of Phytochemicals against the Root-Lesion Nematode *Pratylenchus penetrans*

**DOI:** 10.3390/plants13050726

**Published:** 2024-03-04

**Authors:** Pedro Barbosa, Jorge M. S. Faria, Tomás Cavaco, Ana Cristina Figueiredo, Manuel Mota, Cláudia S. L. Vicente

**Affiliations:** 1MED—Mediterranean Institute for Agriculture, Environment and Development & CHANGE—Global Change and Sustainability Institute, Institute for Advanced Studies and Research, Universidade de Évora, Pólo da Mitra, Ap. 94, 7006-554 Évora, Portugal; pbarbosa@uevora.pt (P.B.); mmota@uevora.pt (M.M.); 2Instituto Nacional de Investigação Agrária e Veterinária (INIAV, I.P.), Quinta do Marquês, 2780-157 Oeiras, Portugal; fariajms@gmail.com (J.M.S.F.); tomasfcavaco@gmail.com (T.C.); 3GREEN-IT Bioresources for Sustainability, Instituto de Tecnologia Química e Biológica, Universidade Nova de Lisboa (ITQB NOVA), Av. da República, 2780-157 Oeiras, Portugal; 4Centro de Estudos do Ambiente e do Mar (CESAM Lisboa), Faculdade de Ciências, Universidade de Lisboa, Biotecnologia Vegetal, DBV, C2, Piso 1, Campo Grande, 1749-016 Lisboa, Portugal; acsf@fc.ul.pt

**Keywords:** pest management, plant secondary metabolites, *Pratylenchus penetrans*, Nematicides

## Abstract

Plant-parasitic nematodes (PPNs) are highly damaging pests responsible for heavy losses in worldwide productivity in a significant number of important plant crops. Common pest management strategies rely on the use of synthetic chemical nematicides, which have led to serious concerns regarding their impacts on human health and the environment. Plant natural products, or phytochemicals, can provide a good source of agents for sustainable control of PPNs, due to their intrinsic characteristics such as higher biodegradability, generally low toxicity for mammals, and lower bioaccumulation in the environment. In this work, the nematicidal activity of 39 phytochemicals was determined against the root-lesion nematode (RLN) *Pratylenchus penetrans* using standard direct and indirect contact methodologies. Overall, the RLN was tolerant to the tested phytochemicals at the highest concentration, 2 mg/mL, seldom reaching full mortality. However, high activities were obtained for benzaldehyde, carvacrol, 3-octanol, and thymol, in comparison to other phytochemicals or the synthetic nematicide oxamyl. These phytochemicals were seen to damage nematode internal tissues but not its cuticle shape. Also, the environmental and (eco)toxicological parameters reported for these compounds suggest lower toxicity and higher safety of use than oxamyl. These compounds appear to be good candidates for the development of biopesticides for a more sustainable pest management strategy.

## 1. Introduction

The root-lesion nematodes (RLNs) of the genus *Pratylenchus* (Nematoda: Pratylenchidae) are migratory endoparasites that affect many agricultural crops with economic importance (such as carrot, coffee, corn, or potato) [1]. The RLN spends most of its life cycle inside plant roots, where it feeds on cortical and stellar cells, but also in soil, feeding on root hairs and epidermal cells [2]. RLN infection typically produces necrotic lesions on infected roots, which begin as necrotic spots on individual cells that may coalesce into larger necrotic lesions as the nematode continues to move and feed within tissues [3]. These lesions weaken the plant, making it more susceptible to other opportunistic soil pathogens such as bacteria and/or fungi [4,5]. Currently, the genus *Pratylenchus* contains 103 species, with an impressive amount of cryptic biodiversity [6]. *Pratylenchus penetrans* (Cobb, 1917) Filipjev and Schuurmans-Stekhoven 1941 is among the most severe RLNs, with more than 400 species of plant hosts [2]. This RLN species reproduces sexually, with females laying single eggs inside plant roots or outside on the rhizosphere. Within the egg, the first juvenile stage (J1) molts to a second-stage juvenile (J2), which hatches and develops into J3 and J4 before reaching adult form. Mobile stages (J2 to adults) can enter and exit roots [2]. *P. penetrans* can reproduce across a wide temperature range, with higher temperatures speeding up the life cycle [7]. Under unfavorable conditions, like the absence of a host or during lower temperatures, *P. penetrans* can enter a developmental dormancy or diapause [8]. If needed, nematodes can even remain in a state of anhydrobiosis [9]. Several studies have shown that the anhydrobiotic state of some *Pratylenchus* species can go up to 21 months under controlled conditions [10,11].

Agricultural practices often employed for RLN control include fallowing, growing cover crops, or crop rotation, although their effects are limited since *P*. *penetrans* has a wide host range [2]. Moreover, only a few agronomic crops have been found to possess moderate resistance to this nematode [12]. Recently, seed meals from species of Brassicaceae (e.g., mustards, cabbages, or broccoli) used as soil amendments have shown a suppressive effect on populations of *P*. *penetrans* [13]. Also, marigold (*Tagetes patula*) is well-known for reducing *P*. *penetrans* populations in field experiments, its effect being persistent when employed as part of crop rotation [14,15]. Plant extracts employed as soil amendments have also shown suppressiveness towards other Pratylenchids [16,17].

For many years, the nematicidal compounds aldicarb, carbofuran, and 1,3-dichloropropene were used indiscriminately in the control of PPNs such as *P. penetrans* [18,19]. With long-term effects on human health and the environment, these compounds of synthetic origin are now forbidden in many countries [20]. The selection of environmentally safe and effective ways to control PPN, particularly RLNs, thus becomes imperative. The search for plant-derived compounds has gained increasing importance due to the ecological advantages they provide in comparison to synthetic chemicals, e.g., higher biodegradability, lower toxicity to mammals, and a long history of use [20]. Plants are known to produce more than 200,000 chemical compounds, many of which originate from specialized biosynthetic pathways [21]. Pioneer studies with complex mixtures of phytochemicals, in the form of extracts, essential oils, or fractions, have proven to possess strong insecticidal [22,23,24], bactericidal [25,26,27], or nematicidal activities [28,29,30,31]. The existing literature on the effects of these mixtures against the most economically important PPNs is extensive, e.g., the activity of essential oils against *Bursaphelenchus xylophilus* [29,32] or against root-knot nematodes [30,33]; the activity of tannins against *Meloidogyne incognita* [34,35]; the activity of alkaloids isolated from *Macleaya cordata* against *B. xylophilus* or *M. incognita* [36] or the alkaloids extracted from several plants against *M. incognita* and *Rotylenchulus reniformis* [37]; the activity of pyrrolizidine alkaloids isolated from *Crotalaria* sp. and *Senecio jacobaea* against *M. incognita* and *P. penetrans* [38]; and the activity of isothiocyanates, obtained from their corresponding glucosinolates in *Brassica* species, against *M. incognita* or *M. hapla* [39,40].

The development of sustainable pest management strategies for the control of *P. penetrans* is crucial. In this sense, we evaluated the nematicidal activity of 39 phytochemicals by direct- and indirect-contact bioassays. The selection of these compounds was based on previous work on essential oils of Mediterranean flora [29,31]. Moreover, we also reported and predicted impacts on the environment and human health. The most promising compounds are strong candidates to control RLN populations, be easily applicable, and allow the plant to perform its normal life cycle.

## 2. Results

### 2.1. Direct Nematicidal Activity

To understand the biopesticidal potential of important phytochemicals, direct contact bioassays were performed to determine specific RLN mortality parameters. For the *P. penetrans* isolate used, control mortality, i.e., mortality in the presence of the compound solvent, was 9.5 ± 0.2% (with a 95% confidence interval of 9.1 to 9.9%). Nematicidal activity was bioassayed for 39 pure phytochemical standards, namely, 20 terpenes, 11 phenolic and other shikimic acid pathway related compounds, 1 alkaloid, 6 fatty acid derivatives, and 1 β-keto acids derivative (Table 1). 

In the case of the terpenes, the lowest mortalities were obtained for hydrocarbons (with corrected mortalities that varied between 1.6 ± 0.4 and 8.3 ± 0.9%), followed by monoterpene ketones, which induced corrected mortalities of 26.6 ± 1.6 and 34.4 ± 1.3%. High corrected mortality values were obtained for monoterpene alcohols (showing mortalities that varied between 13.5 ± 1.0 and 70.0 ± 1.0%) and aldehydes (which induced corrected mortalities of 55.8 ± 1.5 and 66.8 ± 1.5%). The phenol-like monoterpenes tested, namely, carvacrol and thymol, showed complete mortality at 2 mg/mL. Overall, the oxygen-containing terpenes were more successful in inducing mortality in the RLN than hydrocarbons. 

Among the compounds resulting from biochemical pathways that lead to phenols biosynthesis, benzenoid, derived from trans-cinnamic acid by carbon chain shortening, benzaldehyde, was the only one to attain full mortality. From the remaining phenolic compounds tested, only the phenylpropanoids trans-anethole and eugenol (with corrected mortalities of 54.1 ± 2.9 and 59.6 ± 1.8%, respectively) and methyl salicylate (76.8 ± 1.5%) induced moderate nematicidal activities. The remaining phenolic acids showed very low activity or were inactive against the RLN. 

The alkaloid caffeine was inactive at the tested concentration. For compounds outside the previously described chemical classes, namely, 1-decanol, 1-dodecanol, 3-octanol, or 1-undecanol, high mortalities were obtained: 68.8 ± 0.9, 99.1 ± 0.4, and 87.6 ± 0.6, respectively, even higher than the nematicidal compound oxamyl, which induced a corrected mortality of 65.6 ± 2.1%. For 1-tridecanol and 2-undecanone, moderate activities were obtained (58.6 ± 1.8 and 63.2 ± 1.3%, respectively), while 2-octyl-1-decanol can be considered inactive, showing an Mc of 3.2 ± 0.5%.

### 2.2. Toxicological Characterization of Nematicidal Phytochemicals

The most active phytochemicals were tested at lower concentrations to determine their toxicity parameters. The half-maximal effective concentration (EC_50_, in mg/mL) and lowest maximal effective concentration (EC_100_, in mg/mL) values were determined for 16 compounds (Table 2).

Almost every compound showed EC_50_ values between 1 and 3 mg/mL; however, four compounds stood out for their low EC_50_ values, namely, benzaldehyde (0.45 ± 0.01 mg/mL), carvacrol (0.48 ± 0.01 mg/mL), 3-octanol (0.68 ± 0.01 mg/mL), and thymol (0.50 ± 0.01 mg/mL) (Table 2, Figure 1a). These compounds showed low EC_100_ values, i.e., concentrations at which the full RLN population was eliminated (1.5 to 1.7 mg/mL) and low ET_50_ values, i.e., the minimum time required to eliminate half of the RLN population (52 to 62 min) (Figure 1b).

For the most successful phytochemicals, micromorphological evaluations were performed (Figure 2). Carvacrol, 3-octanol, and thymol induced physiological changes in *P. penetrans*, namely the nematodes became darker, which interfered with the observation of the esophagus/intestine interface; the internal structures of the anterior region (metacorpus and esophageal glands) were twisted and wrinkled; and the intestine appeared heavily vacuolated. Specifically, the RLNs appeared to die with an extended stylet and a deformed medium bulb when exposed to 3-octanol. For benzaldehyde, no evident changes were observed. Overall, structures such as the stylet or spicules seemed unaffected, as well as the cuticle shape.

### 2.3. Indirect-Contact Bioassays

For the most active compounds, indirect-contact bioassays were performed to assess their potential for fumigation, i.e., their activity when volatilized. The four compounds (carvacrol, thymol, 3-octanol, and benzaldehyde) showed lower activities compared to when tested in direct-contact bioassays and were unable to induce complete mortality at the highest concentration (2 mg/mL) (Table 3). Despite having similar mortalities and ET_50_ values, the reported vapor pressure values, which indicate the evaporation rate of a liquid compound, are higher for benzaldehyde and 3-octanol, suggesting potentially higher success as fumigants.

### 2.4. Toxicity to Mammals 

The risks for human health were estimated by comparing the reported and predicted toxicity values of the most successful compounds to those of the synthetic nematicide oxamyl. The dermal LD_50_ values reported in tests with rats are very high for all analyzed compounds and can be considered safe; however, the reported oral LD_50_ values are more than 300-fold higher for the selected compounds than for oxamyl (Table 4). Predicted oral LD_50_ values show a tendency similar to the reported experimental values. The predicted mutagenicity level was positive for oxamyl and negative for the phytochemicals analyzed, suggesting higher safety in their use.

### 2.5. Potential Environmental Safety

Environmental safety was assessed by applying predictive models for compound distribution, persistence, and removal in environmental compartments. Concerning environmental distribution, the tested compounds showed a relatively high affinity for the water environmental compartment (Appendix A). The nematicide oxamyl was seen to be almost completely retained in the water compartment (97%), followed by benzaldehyde (65%), 3-octanol (42%), and finally carvacrol and thymol (23% each) (Table 5). Due to their chemical properties, benzaldehyde and 3-octanol additionally showed good affinity to the air environmental compartment (34 and 75%, respectively), while carvacrol and thymol showed high affinity to the soil environmental compartment (73 and 74%, respectively). Thus, the main predicted environmental compartments to be influenced are water for oxamyl, water and air for benzaldehyde and 3-octanol, and water and soil for carvacrol and thymol. Environmental persistence was predicted to be higher for oxamyl than for 3-octanol (4.6-fold), benzaldehyde (3.3-fold), carvacrol (2.7-fold), or thymol (2.5-fold). In aquatic environments, compound volatilization was predicted to be much higher for the phytochemicals analyzed than for oxamyl, most likely due to its high affinity to the water environmental compartment and high solubility in water (148 g/L). The half-life values for benzaldehyde and 3-octanol were predicted to be 24 and 23 h, respectively in the river model, and 344 and 343 h, respectively, in the lake model, while the half-life values for carvacrol and thymol were higher, namely, 444 and 206 h, respectively, in the river model, and 4949 and 2347 h, respectively, in the lake model. In comparison, predicted values were much higher for oxamyl (Table 5). Predicted amounts for removal in wastewater treatment are low for the compounds analyzed, with the lowest percentage for oxamyl (2%), followed by benzaldehyde (3%), 3-octanol (4%), and carvacrol and thymol (6%). 

The toxicity of the tested phytochemicals on aquatic organisms was additionally assessed by reviewing the available data on online databases [41,42,43]. In comparison to oxamyl, the phytochemicals showed higher concentration thresholds (EC_50_ values in mg/L) (Table 6). For the model species of invertebrates, fish, and algae, 3-octanol was the least toxic, with high threshold values, followed by benzaldehyde, carvacrol, and thymol, apart from algae, where thymol showed the second least toxic EC_50_ (12 mg/L).

## 3. Discussion

The successful development of biopesticides for application in sustainable agricultural systems is an important step in the establishment of integrated pest management approaches, serving as mediators of food security to meet the increasing demand for food supplies to support a rapidly growing human population [47]. The combination of phytochemical screening with activity-guided optimization of chemical properties is generally relied upon to pinpoint nematicidal chemical structures. Against the RLN, *P. penetrans*, the present work analyzed the nematotoxic effect of 39 phytochemicals as biopesticides in direct- and indirect-contact bioassays. From these, 24 compounds were tested for the first time against this RLN, and 10 phytochemicals showed stronger activities than oxamyl (>66%), a commonly used commercial nematicide. Overall, a substantial tolerance of *P. penetrans* to the tested compounds was seen, with very few phytochemicals achieving complete mortality at the highest tested concentration. This high endurance has been seen before in direct-contact bioassays with other phytochemical classes. For example, when bioassayed with three acetylene compounds extracted from *Coreopsis lanceolata*, high nematicidal activities were reported for *Bursaphelenchus xylophilus* and *Caenorhabditis elegans* but not for *P. penetrans* [48]. The same observation was reported when assaying several phenolic compounds, terpenes, and alkaloids against the plant-parasitic nematodes *Radopholus similis*, *P. penetrans*, and *Meloidogyne incognita*, with strong inhibition of nematode motility and egg hatching for *R. similis* and *M. incognita* but not for *P. penetrans* [49]. 

In the present study, low activities were observed mainly for terpene hydrocarbons, phenylpropanoids, and phenolic acids. Strong activities were observed for the alcohols 1-decanol, 3-octanol, and 1-undecanol, and for the benzenoid Benzaldehyde; moderate activities were observed for the aldehyde citral, the alcohols 1-dodecanol and geraniol, the ester methyl salicylate, and the ketone 2-undecanone; and weak activities were observed for the alcohols isopulegol, α-terpineol, terpinen-4-ol, and 1-tridecanol, the aldehyde citronellal, and the phenols *trans*-anethole and eugenol. Overall, oxygen-containing terpenic compounds showed higher activities than terpene hydrocarbons, without this electronegative element. This tendency has been reported before for other plant-parasitic nematodes, e.g., the pinewood nematode [31] or the root-knot nematode [30]. 

Against the RLN *P. penetrans*, there is a limited amount of literature on nematicidal phytochemicals. In a similar study, several monoterpenes were tested at 0.25 mg/mL, but only the alcohol citronellol showed strong activity, while carvacrol, thymol, geraniol, and the nematicide oxamyl showed moderate activities [50]. Among the less active compounds were citronellal, eugenol, menthol, limonene, 1.8-cineole, linalool, or α-terpineol. 

For the phenolics tested in the present study, only the benzenoid benzaldehyde, methyl salicylate, and the phenylpropanoids showed high activities when compared to the remaining phenolic compounds. In similar studies, many of the phenolic compounds tested were also ineffective. For example, no substantial influence was reported for chemotactic, motility inhibitory, or anti-hatching bioassays with *P. penetrans* using the phenolic compounds coumaric, chlorogenic, salicylic, caffeic, or ferulic acids [49]. 

In the present study, the alkaloid caffeine was inactive against the RLN *P. penetrans*; however, other studies have shown that the activity of alkaloids might be imposed at higher concentrations and over longer periods of exposure. For example, against *P. penetrans*, only a few pyrrolizidine alkaloids were reported to be active and at higher concentrations than those used in the present study [38]. 

The most active compounds, with corrected mortalities of 100%, were benzaldehyde, carvacrol, 3-octanol, and thymol. These were further analyzed to determine their toxicological parameters and biopesticidal potential. The isomers carvacrol and thymol showed the lowest EC_100_ and ET_50_ values, suggesting fast and strong activity at low concentrations. However, benzaldehyde showed the lowest EC_50_, which indicates that its activity is stronger but progressive, taking longer to reach full mortality. When analyzed microscopically, the changes in morphology were also different for benzaldehyde. While carvacrol, 3-octanol, and thymol induced degradation of internal tissues that led to characteristic vacuolization (probably lipidic in nature), benzaldehyde acted with no apparent morphological disturbance. 

Differential effects due to the application of natural compounds on plant-parasitic nematode ultrastructures are known to occur. For example, *Meloidogyne incognita* J2 were assayed with acetic acid, the ketone 2-undecanone, and the aldehyde *trans*-2-decenal [51]. At the highest concentration, 0.5 mg/mL, the compounds induced intensive vacuolization. However, while acetic acid mainly influenced the cuticle and degenerated the nuclei of pseudocoel cells, *trans*-2-decenal and 2-undecanone showed no negative influence on cuticle structure or somatic muscles but degenerated the pseudocoel cells, with *trans*-2-decenal causing malformation of somatic muscles. 

To understand their potential for fumigation, indirect-contact bioassays were performed with the most active compounds for the first time against *P. penetrans*. Unlike in direct-contact bioassays, these compounds were unable to reach complete mortality at the highest tested concentration. Corrected mortality varied between 54 and 64%, while ET_50_ values varied between 24 and 27 h. However, due to their higher reported vapor pressure values, we proposed that benzaldehyde and 3-octanol might have a higher potential for fumigation than carvacrol and thymol. 

In another study that analyzed the indirect contact activity of several monoterpenes at 0.25 µL (or mg)/mL against *P. brachyurus,* benzaldehyde and geraniol showed moderate activities, while thymol, carvacrol, and 2-octanol were less active [52]. The incorporation of plants that produce bioactive volatiles in soil is nowadays an effective and sustainable way to control PPN [53,54]. Control of nematode population is attributed to a process known as bio-fumigation [55]. Understanding the mechanisms involved in this process can improve the efficiency of nematode control. In addition to their nematicidal effect, volatiles can attract natural enemies and act as an activation signal for resistance-related genes [56]. Another characteristic of using volatiles for nematode control in soil is their solubility in water, which explains their long-distance effects [57]. In the present study, indirect-contact bioassays were performed to infer the ability of the tested compounds to maintain nematicidal activity as a volatile, mimicking the bio-fumigation effect. Crops like oregano (*Origanum vulgare* L.), fennel (*Foeniculum vulgare* L.), or thyme (*Thymus vulgaris* L.) are known to produce some of the most active compounds analyzed in the present study. In addition to their agronomic value per se, they could be employed in traditional crop rotation or incorporated into the soil. For example, marigold (*Tagetes* spp.) is a good example of effectiveness against the RLN [15,58]. For plants that naturally produce nematotoxic compounds, and considering the most recent research, the possibility to enhance plant defenses, and thus, the availability of plant secondary metabolites, opens avenues in agricultural biotechnology [59]. 

The different chemical properties of the phytochemicals analyzed in the present study can influence how they are used but also how they spread to the environment. While oxamyl is predicted to be mainly retained in water, with high persistence in water bodies, the analyzed phytochemicals are predicted to disperse to water but also to air (benzaldehyde and 3-octanol) and soil (carvacrol and thymol) biological compartments. This behavior, allied with lower reported toxicity to aquatic organisms, suggests that resorting to these compounds for the creation of bionematicides can provide sustainable alternatives to commonly used pesticides. Additionally, considering their low persistence in the environment and lower reported and predicted acute toxicities to mammals, we propose that, compared to commonly used nematicides, their use can be safer for the farmer and the consumer, since some of these phytochemicals are already approved as flavoring agents for food [41]. However, further studies are needed to assess the biodegradability of the most interesting phytochemicals (i.e., benzaldehyde, which upon contact with air, can be degraded into other toxic compounds). 

## 4. Materials and Methods

### 4.1. Chemicals

Pure chemical standards of selected plant secondary metabolites were acquired from commercial sources. *trans*-anethole (99%), benzaldehyde (≥99.5%), caffeic acid (98%), caffeine (pure), δ-3-carene (90%), carvacrol (98%), *trans*-β-caryophyllene (≥98.5%), catechin (98%), citral (95%), citronellal (96%), *p*-coumaric acid (98%), *p*-cymene (99%), 1-decanol (≥98%), 1-dodecanol (≥98%), 1-tridecanol (97%), 1-undecanol (99%), eugenol (99%), geraniol (98%), isopulegol (99%), limonene (97%), linalool (98%), methyl salicylate (≥99%), menthol (99%), 2-octyl-1-decanol (97%), piperitone (analytical standard), α-pinene (≥99%), β-pinene (analytical standard), 3-octanol (analytical standard), pulegone (96%), quercetin (≥95%), sabinene (75%), α-terpineol (≥96%), terpinen-4-ol (≥95%), γ-terpinene (97%), thymol (99%), and 2-undecanone (99%) were acquired from Sigma-Aldrich (Lisboa, Portugal); ferulic acid (for research only), gallic acid (for research only), and gentisic acid (for research only) were acquired from Extrasynthèse (Genay, France). All compounds were diluted in acetone (99.8%, Carl Roth GmbH + Co. KG.Portugal) to an initial concentration of 200 mg/mL. Phytochemical stock solutions were stored at −20 °C until used. The commercially available nematicide oxamyl (AFROMYL^®^, Epagro) was also tested at 2 mg/mL in water. 

### 4.2. Nematode Culture and Maintenance

The *P. penetrans* isolate A44L4 was collected in 2010 at a potato field (Coimbra, central Portugal) and kindly provided by NematoLab (Coimbra, University of Coimbra) [60]. RLNs were routinely multiplied in carrot disks according to standardized methodologies [61]. In brief, carrots (*Daucus carota* var. Nantes) were thoroughly washed under running water to remove soil and debris, washed with distilled water, and then surface-sterilized in a laminar flow cabinet by dipping in 96% ethanol (*v*/*v*) (LabChem, Portugal), followed by flame sterilization. The peel was carefully removed, the upper and lower portions were discarded, and the center portion cut into 0.5 cm thick sections. These sections were then placed on sterile Petri dishes and subjected to UV radiation for 2 h. Following this, the sections were stored at 25 °C in darkness for ca. one week, and ca. 60 sterilized RLNs were added to each section and stored for 3 months under the previously described conditions. Afterwards, RLNs were extracted from carrot disks for 24 h with distilled water supplemented with carbenicillin and kanamycin at 50 μg/mL each. Finally, a suspension of 50 and 75 mixed-stage RLN per 100 μL was prepared to be used in the bioassays.

### 4.3. Direct-Contact Bioassays

The nematicidal activity of each phytochemical and the commercial nematicide oxamyl was assessed by direct-contact bioassays [15,16]. Briefly, 1 μL of phytochemical stock solution was added to 99 μL of a mixed-stage nematode suspension (for a final concentration of 2 mg/mL) per well in a 96-well microtiter plate (Carl Roth GmbH & Co. KG, Karlsruhe, Germany). The plates were then covered and stored in darkness at 25 °C. After 24 h, dead and live nematodes were counted using a binocular microscope Olympus SZX-12 (10×) (Olympus Corporation, Tokyo, Japan). Recovery tests were performed by immediately transferring RLNs to distilled water on a Petri dish for 2 h and then reassessing mortality. Nematodes were considered dead when no movement was detected even after physical prodding with a needle. Three independent biological trials were performed, each with five repetitions in similar conditions. Control bioassays were performed by adding 1 μL of acetone (99.8%). 

To understand the time required for phytochemical activity, RLN mortality was determined at different exposure times. Bioassays were performed as previously described, and mortality was assessed at 5, 15, 30, and 60 min, and 3, 6, 12, and 18 h after exposure to the 2 mg/mL compound solution. After each time-point, recovery was tested as described above. Three independent biological trials were performed, each with five repetitions under similar conditions.

### 4.4. Indirect-Contact Bioassays

To assess the potential for fumigant nematicidal activity, the most active phytochemicals were tested by indirect-contact bioassays. For this effect, for each phytochemical, a 96-well microtiter plate was prepared as follows: a small rectangle (40 mm × 5 mm) of filter paper (Rotilabo^®^, Typ 11A, cellulose, ⌀ membrane 55 mm) was bridged in-between adjacent wells containing the nematode suspension (99 μL). Each phytochemical solution (1 μL at 2 mg/mL) was added to the filter paper, standing approximately 1.5 cm above the nematodes (i.e., height of the well). Plates were covered, sealed with parafilm to minimize compound volatilization, and stored in the dark at 25 °C. After 24 h, dead and live nematodes were counted as described above. To avoid cross effects, only one well was used per plate. Five independent biological trials were performed, with five replicates under similar conditions. 

### 4.5. Microscopic Analysis of RLN Body Structure

To evaluate whether the most promising phytochemicals were causing structural damage, 24 h-treated nematodes were mounted in distilled water, observed directly under an inverted microscope Leica DMi1 (Leica Microsystems AG, Wetzlar, Germany), and documented with a Flexicam C1.

### 4.6. Toxicological and Ecotoxicological Parameters 

Toxicity and ecotoxicity parameters were assessed by reviewing data on the reported (eco)toxicological parameters of the most active nematicidal phytochemicals and the synthetic nematicide oxamyl. These data were retrieved from PubChem [41], PPDB: the Pesticide Properties Database [43], and ECHA, the European Chemicals Agency [42]. For the predicted toxicological parameters, T.E.S.T. software, developed by the U.S. Environmental Protection Agency, was used [44] (Appendix A). 

The predicted environmental distribution of the phytochemicals was compared to that of the synthetic nematicide oxamyl by determining predicted environmental distribution (PED) percentages through the predictive equilibrium criterion model suggested by Mackay et al. [45], using the freely available Level I Mackay Fugacity Model beta version 4.31, Trent University, Canada [62]. The chemical descriptors needed from each compound, namely, molecular mass (g/mol), melting point (°C), vapor pressure (Pa), solubility in water (mg/L), air–water partition coefficient or Henry’s Law constant (Pa.m^3^/mol), *n*-octanol/water partition coefficient (log value of Kow), and soil organic carbon/water partition coefficient (Koc) were retrieved from the PubChem [41], PPDB: the Pesticide Properties Database [43], and ECHA, the European Chemicals Agency [42] online databases.

### 4.7. Data Treatment and Statistical Analysis

Data processing was performed with version 27 of SPSS Statistics software, version 27 (IBM, New York, NY, USA). Statistical significance of the data was determined with one-way ANOVA, and individual means were compared using Tukey’s post hoc test with *p* < 0.05. The Shapiro–Wilk test ensured data normality, and the Browns–Forsythe test was used for homoscedasticity. Corrected mortality (Mc) percentages for each compound, at each concentration and time-point, were determined using the Schneider–Orelli formula [63], M_C_ = (M_T_ − M_0_)/(100 − M_0_), where M_0_ is the mortality % in control and M_T_ is the mortality % in treatments. Nematicidal activity was considered strong for M_C_ above 80%, moderate between 80 and 61%, weak for M_C_ between 60 and 40%, and low or inactive when below 40% [64]. 

Determination of the half-maximal effective concentration (EC50) [65] for each compound was performed with version 2019 of Origin Graphing and Analysis software (OriginLab, Northampton, MA, USA). A nonlinear regression analysis was performed by plotting Mc values against compound concentration and fitting a dose–response log–logistic equation, y = C + (D − C)/1 + exp {b [log (x) − log (EC50)]}, where C and D are the lower and upper limits of the sigmoidal dose–response curve, respectively, b is the slope, and EC50 is the compound concentration that induces a response halfway between the lower and upper limits. To determine the half-maximal effective time (ET50) values, the same procedure was performed, but the time-point values were used instead of compound concentrations.

## 5. Conclusions

*Pratylenchus penetrans* is considered one of the most important RLNs in agriculture. This species is mostly tolerant to common plant-derived compounds used to control other plant parasitic nematodes. The lack of efficient control measures for this RLN emphasizes the need to continue testing novel compounds. In this study, four phytochemicals (thymol, carvacrol, benzaldehyde, and 3-octanol) were identified as strong nematicides against *P. penetrans* in direct and indirect applications. 

## Figures and Tables

**Figure 1 plants-13-00726-f001:**
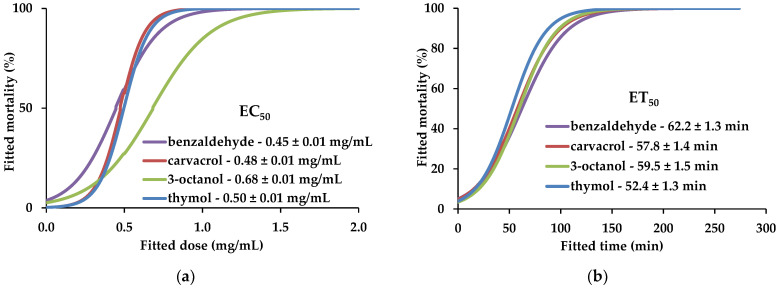
Sigmoidal curves fitted to dose–response, at 24 h (**a**), and time–response, at 2 mg/mL (**b**). Data obtained for benzaldehyde (purple), carvacrol (red), 3-octanol (green), and thymol (blue) in direct-contact bioassays with the *Pratylenchus penetrans*. Half-maximal effective concentration (EC_50_) and half-maximal effective time (ET_50_) values are presented as average ± standard error.

**Figure 2 plants-13-00726-f002:**
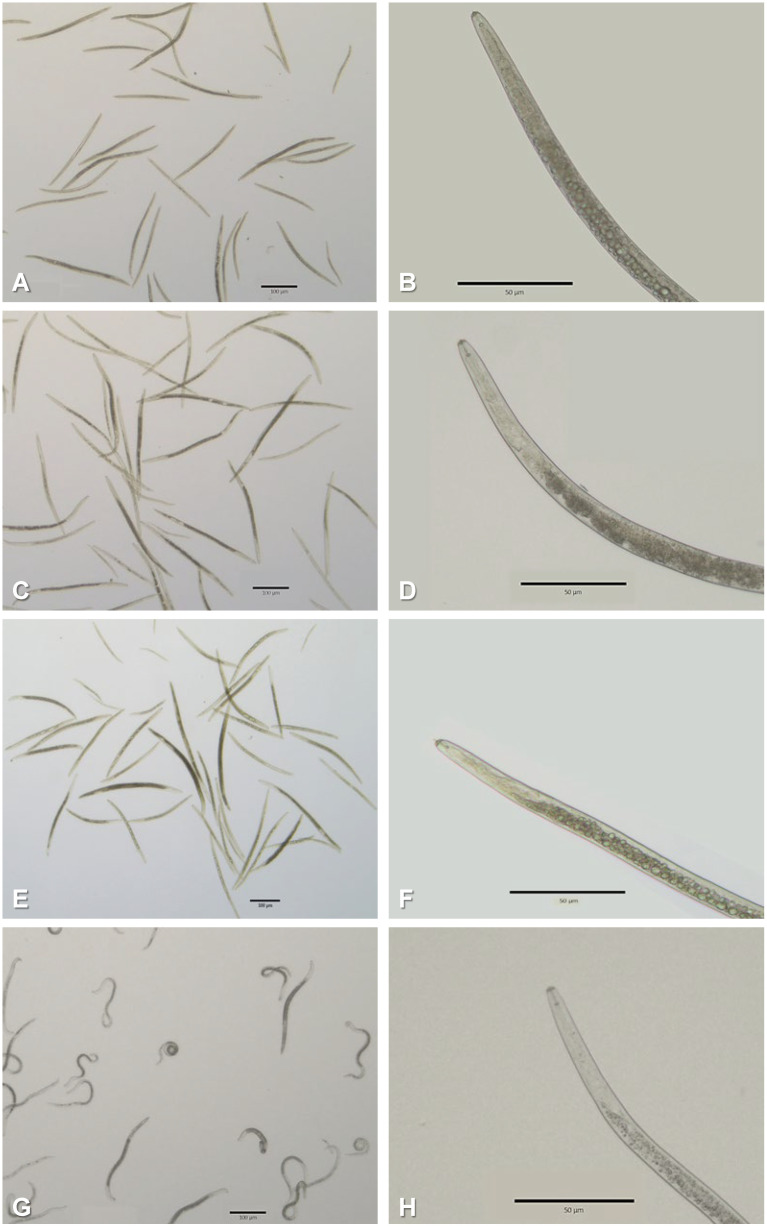
Micrographs of *Pratylenchus penetrans* in direct contact with carvacrol/thymol (**A**,**B**), benzaldehyde (**C**,**D**), 3-octanol (**E**,**F**), and control (**G**,**H**). Root-lesion nematodes were exposed for 24 h to 2 mg/mL of each phytochemical.

**Table 1 plants-13-00726-t001:** Corrected mortality (Mc) induced by the phytochemicals and the synthetic nematicide oxamyl (2 mg/mL) on *Pratylenchus penetrans*, after 24 h of direct-contact bioassays. Compounds are ordered by decreasing Mc within their chemical class.

Compound	Chemical Classification	Mc (%)
	**Terpenes**	
Carvacrol	monoterpene phenol	100.00 ± 0.00
Thymol	monoterpene phenol	100.00 ± 0.00
Geraniol	monoterpene alcohol	70.03 ± 1.04
Citral	monoterpene aldehyde	66.81 ± 1.53
Citronellal	monoterpene aldehyde	55.79 ± 1.54
α-Terpineol	monoterpene alcohol	45.88 ± 1.91
Terpinen-4-ol	monoterpene alcohol	42.09 ± 1.84
Isopulegol	monoterpene alcohol	41.71 ± 2.37
Pulegone	monoterpene ketone	34.43 ± 1.30
Piperitone	monoterpene ketone	26.63 ± 1.63
Linalool	monoterpene alcohol	25.76 ± 1.60
Menthol	monoterpene alcohol	13.48 ± 1.01
Sabinene	monoterpene hydrocarbon	8.31 ± 0.89
*p*-Cymene	monoterpene hydrocarbon	8.14 ± 0.92
Limonene	monoterpene hydrocarbon	7.08 ± 1.16
γ-Terpinene	monoterpene hydrocarbon	6.93 ± 0.83
*trans*-β-Caryophyllene	sesquiterpene hydrocarbon	5.08 ± 0.69
δ-3-Carene	monoterpene hydrocarbon	5.07 ± 0.72
α-Pinene	monoterpene hydrocarbon	1.89 ± 0.56
β-Pinene	monoterpene hydrocarbon	1.64 ± 0.41
	**Phenolics**	
Benzaldehyde	phenylpropanoid/benzenoid	100.00 ± 0.00
Methyl salicylate	salicylate ester	76.69 ± 1.53
Eugenol	phenylpropanoid	59.60 ± 1.83
*trans*-Anethole	phenylpropanoid	54.09 ± 2.85
Gentisic acid	phenolic acid	5.98 ± 1.30
Caffeic acid	phenolic acid	4.63 ± 0.75
Catechin	flavonoid	2.26 ± 0.61
Ferulic acid	phenolic acid	2.06 ± 0.34
Coumaric acid	phenolic acid	1.88 ± 0.37
Gallic acid	phenolic acid	1.44 ± 0.41
Quercetin	flavonoid	1.37 ± 0.34
	**Alkaloids**	
Caffeine	xanthine	1.55 ± 0.39
	**Fatty acids derivatives**	
3-Octanol	fatty alcohol	99.06 ± 0.35
1-Decanol	fatty alcohol	89.44 ± 0.86
1-Undecanol	fatty alcohol	87.64 ± 0.58
1-Dodecanol	fatty alcohol	68.85 ± 0.91
1-Tridecanol	fatty alcohol	58.56 ± 1.79
2-Octyl-1-decanol	fatty alcohol	3.25 ± 0.52
	**β-Keto acids derivative**	
2-Undecanone	methyl ketone	63.25 ± 1.31
	**Synthetic nematicide**	
Oxamyl ^1^	carbamate	65.62 ± 2.11

^1^ The active compound of the pesticide afromyl™.

**Table 2 plants-13-00726-t002:** Half maximal effective concentration (EC_50_, in mg/mL) and lowest maximal effective concentration (EC_100_, in mg/mL) of the phytochemicals against *Pratylenchus penetrans*, obtained by fitting a dose–response sigmoidal curve. EC_50_ values are presented as average ± standard error, and EC_100_ as an average with upper and lower 95% confidence intervals. The goodness of fit for sigmoidal curves was evaluated through adjusted R^2^. Compounds are ordered by increasing EC_50_.

Compounds	EC_50_ (mg/mL)	EC_100_ (mg/mL)	Slope	Goodness of Fit (adj. R^2^)
Benzaldehyde	0.45 ± 0.01	1.71 (1.39–1.83)	3.15 ± 0.20	0.97
Carvacrol	0.48 ± 0.01	1.54 (0.81–1.66)	5.63 ± 0.69	0.97
Thymol	0.50 ± 0.01	1.61 (0.79–1.75)	5.38 ± 0.75	0.97
3-Octanol	0.68 ± 0.01	1.71 (1.61–1.78)	2.32 ± 0.09	0.99
Methyl salicylate	0.96 ± 0.04	-	0.97 ± 0.08	0.86
Citral	1.09 ± 0.06	-	0.56 ± 0.05	0.71
Geraniol	1.09 ± 0.06	-	0.75 ± 0.07	0.77
2-Undecanone	1.22 ± 0.07	-	0.52 ± 0.05	0.70
Eugenol	1.72 ± 0.04	-	0.69 ± 0.03	0.92
*trans*-Anethole	1.78 ± 0.07	-	0.64 ± 0.06	0.77
α-Terpineol	2.09 ± 0.04	-	0.66 ± 0.05	0.91
Terpinen-4-ol	2.13 ± 0.04	-	0.71 ± 0.05	0.93
Isopulegol	2.20 ± 0.05	-	0.68 ± 0.06	0.88
Citronellal	2.34 ± 0.09	-	0.52 ± 0.05	0.82
Pulegone	2.50 ± 0.09	-	0.52 ± 0.05	0.83
Linalool	2.56 ± 0.08	-	0.81 ± 0.09	0.89

**Table 3 plants-13-00726-t003:** Corrected mortality (Mc) (24 h at 2 mg/mL) and half-maximal effective time (ET_50_) values for carvacrol, benzaldehyde, 3-octanol, and thymol obtained from indirect-contact bioassays with the root-lesion nematode *Pratylenchus penetrans*. The goodness of fit for sigmoidal curves was assessed by the adjusted R^2^. Values for vapor pressure (Pa), reported on online databases, are provided for comparison purposes. Compounds are ordered by increasing Mc.

Compounds	Mc (%)	ET_50_ (h)	Goodness of Fit (adj R^2^)	Vapor Pressure (Pa) ^1^
Carvacrol	54.08 ± 1.34	24.07 ± 0.58	0.95	3.1–6.7
Benzaldehyde	60.53 ± 1.77	23.54 ± 1.07	0.85	133
3-Octanol	62.09 ± 2.39	24.68 ± 1.45	0.76	34.1
Thymol	63.73 ± 1.51	26.56 ± 1.66	0.71	2.1

^1^ Values retrieved from online databases, PubChem [41] and ECHA [42].

**Table 4 plants-13-00726-t004:** Experimental acute toxicity (oral and dermal) for mammals (median lethal dose, LD_50_, mg/kg) reported in the freely available PubChem online database [41] and PPDB: the Pesticide Properties Database [43]. Predicted acute oral toxicity and mutagenicity levels obtained with the Toxicity Estimation Software Tool, version 5.1.2 (T.E.S.T.) [44].

Reported LD50 (mg/kg)	Benzaldehyde	Carvacrol	3-Octanol	Thymol	Oxamyl
Oral (Rat)	1300	810	>5000	980	3
Dermal (Rat)	>2000	2700	>5000	>2000	5000
Predicted LD50 (mg/kg) ^1^					
Oral (Rat)	1128	1073	2828	507	8
Mutagenicity level ^2^	Negative	Negative	Negative	Negative	Positive

^1^ Toxicity parameters were estimated using the nearest neighbor method, where the predicted toxicity is estimated through comparison with those of three of the most similar chemicals. ^2^ Ames mutagenicity was estimated according to the bacterial reverse mutation assay performed on *Salmonella typhimurium*.

**Table 5 plants-13-00726-t005:** Predicted environmental distribution (PED, %) in the environmental compartments air, water, soil, and sediments using the Mackay fugacity model [45,46], predicted volatilization from water (using river and lake models) [46], and predicted removal in wastewater treatment obtained from the EPI Suite™-Estimation Program Interface software, version 4.11 [46], based on the experimental chemical properties reported on online databases [41,43], for the phytochemicals compared to oxamyl.

PED	Benzaldehyde	Carvacrol	3-Octanol	Thymol	Oxamyl
Air (%)	34	2	55	1	0
Sediments (%)	0	2	0	2	0
Soil (%)	1	73	3	74	3
Water (%)	65	23	42	23	97
Persistence (h)	343	429	251	464	1150
**Volatilization from water ^1^**					
Model river (half-life in h)	24	444	23	206	4 × 10^6^
Model lake (half-life in h)	344	4949	343	2347	4 × 10^7^
**Removal in wastewater treatment ^2^**					
Total removal (%)	3	6	4	6	2
Biodegradation (%)	0	0	0	0	0
Sludge adsorption (%)	2	6	2	6	2
Release to the Air (%)	1	0	2	0	0

^1^ Obtained using WVOLWIN™ module for the estimates of rate of volatilization of a chemical from rivers and lakes; ^2^ obtained using STPWIN™ module to predict the removal of a chemical in a typical activated sludge-based sewage treatment plant [46].

**Table 6 plants-13-00726-t006:** Reported half-maximal effective dose (EC_50_, in mg/L) values for invertebrates, fish, and algae of the phytochemicals compared to oxamyl. Values were retrieved from online databases [41,42,43].

Reported EC50 (mg/L)	Benzaldehyde	Carvacrol	3-Octanol	Thymol	Oxamyl
Invertebrates (48 h) ^1^	20	6	185	5	0.3
Fish (96 h) ^2^	8	6	11 ^4^	5	3
Algae (96 h) ^3^	8	4	114	12	1

^1^ Values reported for the model *Daphnia magna*; ^2^ values reported for *Pimephales promelas*, with benzaldehyde and 3-octanol, for *Brachydanio rerio*, with carvacrol, for *Oryzias latipes*, with thymol, and for ^3^
*Oncorhynchus mykiss*, with oxamyl; ^4^ value was predicted in T.E.S.T. software, version 5.1.2 [44], based on the nearest neighbor method using the model fish *Pimephales promelas*.

## Data Availability

The raw data supporting the findings of this study are available from the corresponding author (Cláudia S. L. Vicente) upon reasonable request.

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
