# Peer review of "Nematicidal Activity of Phytochemicals against the Root-Lesion Nematode Pratylenchus penetrans"

_plants, 2024, doi:10.3390/plants13050726_

Round 1

Reviewer 1 Report

Comments and Suggestions for Authors

ID plants-2895597

Article: Nematicidal activity of phytochemicals against the root-lesion nematode Pratylenchus penetrans

The aim was evaluated the nematicidal activity for 39 pure phytochemical standards by direct- and indirect-contact bioassays.

Comments

The work is justified and substantiated, although I find similarity in other published works. The experimental part is correctly described, as is the discussion, but the conclusion should be reformulated

Please suggest the authors review the novelty regarding benzaldehyde, the effect of this compound is well known (for example: https://doi.org/10.1016/j.soilbio.2007.05.011, in this work is reported: “Here, we identified the following 14 bacterial VOCs with nematicidal effects: phenol, 2-octanol, benzaldehyde, benzeneacetaldehyde, decanal, 2-nonanone, 2-undecanone, cyclohexene, phenyl ethanone, dimethyl disulfide, terpineol, nonane, propanone and benzeneethanol” Gu et al., 2007)

I ask, is the conclusion really novel?,

Author Response

The authors are grateful for the suggestion made by the reviewer. Our conclusions have been improved. The authors note that benzaldehyde has been previously assayed as a nematicidal agent, however we detail for the first time its effectiveness against Pratylenchus penetrans, a soil-dwelling plant parasitic nematode, while previously it was only described for Bursaphelenchus xylophilus, a parasite of pine shoots, and Panagrellus redivivus, a free-living nematode. The novelty of our results lies in its potential as a fumigant, a class of nematicides generally mostly used in the soil. 

Reviewer 2 Report

Comments and Suggestions for Authors

The paper “Nematicidal activity of phytochemicals against the root-lesion nematode Pratylenchus penetrans” describes the study of nematicidal activity of 39 phytochemicals for the root-lesion nematode. The authors have found that high activities were obtained for benzaldehyde, carvacrol, 3-octanol and thymol, in comparison to the other phytochemicals or the synthetic nematicide oxamyl. These phytochemicals were seen damaging nematode internal tissues but not its cuticle shape. Of course these data are important in the development of biopesticides. At the same time the authors don’t take into account such parameter as chemical stability of possible biopesticide. For example, benzaldehyde could be oxidized by air to form benzoyl peroxide (explosive compound) and benzoic acid (that could have or not have nematicidal activity, and have own environmental and (eco)toxicological parameters). These factors should be also mentioned before making a summary of  “good candidates for the development of biopesticides”. The authors should also explain the reasons of choosing exactly these 39 phytochemicals for their research.

Author Response

The authors are grateful for the suggestion made by the reviewer. Considering compounds’ stability, the authors added a sentence in the discussion - L351-353 “However, further studies are needed in terms of biodegradability (i.e., benzaldehyde in contact with air can be degraded in other toxic compounds)”.

The 39 phytochemicals are the major constituents of the essential oils on Mediterranean flora, previously studied by the authors (Barbosa et al., 2010, Faria et al., 2022). A sentence was added in L93-94.

Reviewer 3 Report

Comments and Suggestions for Authors

The authors report the study of nematicidal activities of 39 phytochemicals with the environmental and (eco)toxicological estimated by computer programs or taken from literature.
Methodology
1. Why did authors dilute compounds in acetone? How does acetone affect nematode? Why did they not perform a blind control with acetone only?
2. Why Table 7 is a part of Material and Methods?
3. Data for Vapor pressure are repeated in the Table 3 and 7.
Spelling errors
• line 99: 9.5±0.2 introduce space before and after ± as in Table 1 and 2, and uniform through the entire manuscript,
• Table 7: (Pa.m3/mol)

Comments on the Quality of English Language

Minor editing of English language required

Author Response

The authors are grateful for the questions/corrections made by the reviewer.

  • Regarding the 1st question, the solving agent acetone was selected based on the author's previous studies (Barbosa et al., 2012). Acetone was also tested as a negative control in the bioassays (reference in the text, L400-401). The mortality % for acetone (alone) was 9.4%. This value was considered in the calculation of corrected mortality.
  • The authors agree with the reviewer's question, and moved Table 7 for supplementary table 1. The vapor pressure is now presented only in Supp Table 1.
  • A space between the ± was added in the entire manuscript.
  • The SI unit of Henry’s Law Constant was corrected for m3.Pa/mol.

Round 2

Reviewer 1 Report

Comments and Suggestions for Authors

Accepted